# Nanofibrous Membrane with Encapsulated Glucose Oxidase for Self-Sustained Antimicrobial Applications

**DOI:** 10.3390/membranes11120997

**Published:** 2021-12-20

**Authors:** Fernaldy Leonarta, Cheng-Kang Lee

**Affiliations:** Department of Chemical Engineering, National Taiwan University of Science and Technology, Taipei 106, Taiwan; f.leonarta@gmail.com

**Keywords:** glucose oxidase, hydrogen peroxide, glucose, antimicrobial enzyme, antimicrobial activity, encapsulation, polyvinyl alcohol, simultaneous electrospinning

## Abstract

Polyvinyl alcohol (PVA) nanofibrous membrane, consisting of separately encapsulated glucose oxidase (GOx) and glucose (Glu) nanofibers, was prepared via simultaneously electrospinning PVA/GOx and PVA/Glu dopes. The as-prepared pristine membrane could self-sustainably generate hydrogen peroxide (H_2_O_2_) only in contact with an aqueous solution. The H_2_O_2_ production level was well maintained even after storing the dry membrane at room temperature for 7 days. Cross-linking the membrane via reaction with glutaraldehyde (GA) vapor could not only prevent the nanofibrous membrane from dissolving in water but also prolonged the release of H_2_O_2_. The sustained release of H_2_O_2_ from the membrane achieved antimicrobial capability equivalent to that of 1% H_2_O_2_ against both *Escherichia coli* and *Staphylococcus aureus*. Gram(+) *S. aureus* cells were more susceptible to H_2_O_2_ than Gram(−) *E. coli* and >99% of *S. aureus* were killed after 1 h incubation with the membrane. Pristine and GA-crosslinked nanofibrous membrane with in situ production of H_2_O_2_ were self-sterilized in which no microorganism contamination on the membrane could be detected after 2 weeks incubation on an agar plate. The GOx/Glu membrane may find potential application as versatile antimicrobial materials in the field of biomedicine, in the food and health industries, and especially challenges related to wound healing in diabetic patients.

## 1. Introduction

The investigation of antimicrobial enzymes applications has been growing rapidly in various industries, such as healthcare, food, and biomedical. The antimicrobial activities resulted from these enzymes involving direct attack and lysing or catalyzing reactions to release disinfection compounds that inactivate or kill the microorganisms [1,2,3]. Glucose oxidase (GOx) is one of the widely used oxidoreductase enzymes with potent antimicrobial capability [4]. It exerts its antimicrobial activity by the in situ generation of reactive molecules via catalyzing the reaction between glucose and oxygen, producing D-gluconic acid with concurrent production of hydrogen peroxide (H_2_O_2_) (Figure 1) [5]. The produced hydrogen peroxide is well-known for its disinfection capability, as DNA, proteins, membrane lipids of bacterial cells could be damaged due to oxidation that leads to their death [6]. In terms of effectivity, H_2_O_2_ is believed to be better against Gram-positive bacteria than that of Gram-negative bacteria [7]. Moreover, the pH-lowering effect resulting from produced D-gluconic acid accumulation in the system may also inhibit the growth of a handful of organisms [8].

In addition to its broad antimicrobial applications, GOx also has an important biomedical application for detecting as well as treatment [9] of blood sugar level for diabetic patients based on its reaction with glucose to generate H_2_O_2_, which could be easily determined via amperometric methods [10,11,12,13,14]. Nevertheless, the widespread applications of this GOx-based glucose biosensor are limited due to its cost, instability, and reusability [15]. These major challenges could actually be overcome through selection of a suitable matrix and technique for immobilization and encapsulation of GOx. In recent times, electrospinning has become an often-used enzyme immobilization method. Through this method, the enzyme is encapsulated in various polymeric nanofibers [16] which structures provide good immobilization yield and outstanding thermal stabilization effect [17]. Other than that, the high specific surface area makes it more efficient for the encapsulated enzyme to carry out catalytic reactions with substrate diffuses in from surrounding environment [18,19]. In addition, various electrospun nanofibers have recently been studied for wound dressing applications because the nanofibrous scaffold can provide high porosity, good biocompatibility, which facilitate cell respiration and infiltration for wound tissue regeneration. Furthermore, various nanoparticles have been encapsulated to confer antimicrobial activity on electrospun nanofibers to protect the wound from pathogen infection [20,21,22,23,24,25,26,27,28,29].

Polyvinyl alcohol (PVA) is one of the most easily electrospinable polymers that offers unique characteristics such as excellent gel-forming properties and good film-matrix forming ability. Thus, PVA had been widely exploited for drug delivery and enzyme encapsulation [30,31,32,33,34]. Additionally, PVA is also known for its high biocompatibility and hydrophilicity, ease of processing, non-toxic, and biodegradable nature, confirming its ability to work as a matrix in immobilizing and stabilizing enzymes [35,36]. 

In this work, PVA nanofibrous membranes were prepared via simultaneous electrospinning (co-electrospun) of two different dopes (Figure 1), of which one containing GOx enzyme, and the other, glucose (Glu) as its substrate. Hence, whenever in contact with an aqueous solution, the membrane will be able to provide H_2_O_2_ as an antimicrobial agent in both the presence and absence of glucose in the surrounding environment. To prevent PVA nanofibrous membrane from water-dissolving [37] that leads to the loss of the encapsulated GOx, glutaraldehyde (GA) vapor crosslinking of PVA was implemented [38]. The GA-crosslinked PVA membrane will significantly reduce the diffusion of encapsulated Glu to reach the GOx encapsulated in the other separated nanofibers in an attempt to produce H_2_O_2_. As a consequence, the release of H_2_O_2_ from the GA-crosslinked PVA membrane will be sustained for a longer time as compared to the pristine membrane. To the best of our knowledge, simultaneous electrospinning of PVA/GOx and PVA/Glu to produce a membrane capable of self-sustained generation of H_2_O_2_ for antimicrobial applications has never been reported previously. On the other hand, the membrane with the capability of self-generation H_2_O_2_ also allows supplementary addition of glucose from surrounding to accelerate the disinfection rate through rapid and enhanced H_2_O_2_ generation. Most importantly, this antimicrobial nanofibrous membrane could be a potential wound healing material, especially for diabetic patients since high blood sugar levels will benefit the generation of antimicrobial H_2_O_2_.

## 2. Materials and Methods

### 2.1. Materials

Polyvinyl alcohol (PVA, degree of polymerization = 2000; Mw = 81,000–96,000; hydrolysis 98.5–99.4%, Hanawa Chemical Pure, Osaka, Japan), glutaraldehyde (GA, 25% aqueous solution, Alfa Aesar, Lancashire, England), Glucose Oxidase (GOx,“Amano” NA 233 U/mg, Amano Enzyme Inc., Glucose (α-D(+)-glucose, 99% ACROS-Organics), Monosodium phosphate monohydrate (H_2_NaPO_4_.H_2_O, 99% ACROS-Organics); Disodium Phosphate heptahydrate (HNa_2_PO4.7H_2_O, 99% ACROS-Organics; Hydrogen peroxide (H_2_O_2_, 35 wt% aqueous solution), 3,3′,5,5′-Tetramethylbenzidine, 99+%, (TMB, 99% Thermo Scientific ACROS-Organics, Antwerp, Belgium), N-(carboxymethylaminocarbonyl)-4,4′-bis(dimethylamino)-diphenylamine sodium salt (DA-64, FUJIFILM Wako Chemicals Europe Gmb–Lab Chem, Neuss, Germany), Peroxidase from horseradish (type I, essentially salt-free, lyophilized powder, ≥50 units/mg solid, SIGMA, Marlborough, MA, USA), Sodium acetate anhydrous (98.5%, DUKSAN, Ansan-si, Korea). All Chemicals were used as received.

### 2.2. GOx Encapsulation: Simultaneous Electrospinning of PVA/GOx and PVA/Glucose

PVA (10%) solution was prepared by dissolving PVA powder in phosphate buffer (10 mM, pH 6.8) at 80 °C with vigorous stirring for 3 h. After the solution was cooled down to room temperature, 4 mL of this solution was mixed with 1 mL of 2% GOx enzyme dissolved in phosphate buffer (10 mM, pH 6.8) to give a 4:1 volume ratio of PVA/GOx. Similarly, three sets of 4 mL of PVA solutions were mixed with 1 mL of each three different Glu concentrations (1, 3, and 5 mg/mL) dissolved in phosphate buffer (10 mM, pH 6.8) to give each a 4:1 volume ratio of PVA/Glu. The PVA/GOx and PVA/Glu solutions are loaded into two separate 5 mL syringes fitted with 21G blunt end stainless-steel needles with 0.5 mm inner diameter. Simultaneous electrospinning of the solutions was carried out in a Nanofiber Electro-Spinning Unit (JYI GOANG ENTERPRISE Co., Ltd, New Taipei City, Taiwan) fitted with a high voltage supply (COSMI SC-PME50, New Taipei City, Taiwan) by charging the solutions at 15.5 kV–18.5 kV and the distance between the needle tips and the rotating drum collector (30 rpm) was set to be 110 mm. The two solutions (PVA/GOx and PVA/Glu) were dispensed simultaneously at a flow rate of 9 μL/min. Samples were first placed in an oven at 45 °C for around 30 min to 1 h for complete drying before experiments.

### 2.3. GA Vapor Cross-Linking

The cross-linking procedure was carried out with GA vapor in a sealed vessel. The co-electrospun GOx/Glu nanofibrous membrane was exposed to GA vapor prepared with 2 M GA solution containing 1 M HCl as a catalyst for cross-linking reaction in a 3:1 volume ratio GA:HCl for 12 h.

### 2.4. Characterization

The physical morphology of the co-electrospun GOx/Glu nanofibrous membrane and the cross-linked one was characterized by a field-emission scanning electron microscope (FE-SEM, JEOL JSM-6500 F, Taipei, Taiwan) after sputter coated with platinum for 80 s. The average nanofibers diameter distribution was measured by analyzing the SEM image using ImageJ software.

### 2.5. Hydrogen Peroxide Generation Detection and Sustainability Tests

GOx and Glu co-electrospun nanofibrous membrane was tested for its H_2_O_2_ generation capacity using two different colorimetric assays. Qualitatively, the generation of H_2_O_2_ was determined by the reaction with DA-64 and HRP to produce a green color. As for the H_2_O_2_ production sustainability, three different nanofibrous membranes (PVA only, PVA/GOX, and co-electrospun PVA (GOx/Glu)) without UV-sterilization pretreatment were placed on the surface of agar plate containing glucose for one week. TMB assay using 3,3′,5,5′-tetramethylbenzidine dye was employed to quantify the H_2_O_2_ concentration generated by the membrane. Briefly, reaction solution prepared by mixing sodium acetate buffer (0.1 M, pH 4.5), TMB solution (0.025 M in acetate buffer), and HRP (60 U/mL) with a volumetric ratio of 85:10:5. H_2_O_2_ solutions of different concentrations were added to the reaction solution for establishing a calibration curve. For the study of sustainable H_2_O_2_ production from the membrane, membrane samples (1 cm × 1 cm; ± 4 mg) were incubated in 20 mL DI water, and an aliquot of the sample solution (volume of reaction solution: sample solution = 10:1) was taken for H_2_O_2_ determination at different time. The stability of the membrane for H_2_O_2_ generation was investigated by storing the membranes in dry conditions at room temperature. At different times, several pieces of stored membranes were taken to incubate in DI water for 4 h and the H_2_O_2_ concentration in the water solution was then measured.

### 2.6. Antimicrobial Activity of the GOx Encapsulated PVA Nanofibrous Mat

Gram-negative (*Escherichia coli*) and Gram-positive (*Staphylococcus aureus*) were taken as model strains to examine the antimicrobial activity of the as-electrospun nanofibrous mat. *E. coli* was cultured in LB nutrient culture media and *S. aureus* was cultured in TSB culture media and both cultures were incubated with a shaker for 24 h at 37 °C. The antimicrobial activity of the nanofibrous mat was tested using the inhibition zone method where the bacterial concentration of OD_600_ = 0.5 inside PBS buffer was then plated on the agar plates. To determine the effect of glucose, four different agars (LB, LB with glucose supplement, TSB, TSB without glucose supplement) were used for the two bacteria. The nanofibrous membrane was then plated onto the agar plates and incubated for 24 h at 37 °C. Alternatively, an antimicrobial test via CFU/mL (viable cell counting) was made in order to ascertain the antimicrobial activity of the membranes quantitatively. In a 96-well plate, small cut-out pieces of the membranes (approx. 0.5 cm × 0.5 cm) were placed in six different wells for each bacteria and 0.1 mL of each bacterial solution was transferred into the wells. Six wells were used to measure three different contact times, which were 30 min, 1 h, and 2 h for each bacterial solution. After reaching the supposed contact times, the membranes were taken out of the solution, and serial dilution was conducted in PBS buffer in an attempt to reach countable amounts of bacterial colonies. One hundred microliters of this solution was then plated on each designated agar plate three times, which were then incubated for the same amount of time to the disk diffusion antimicrobial step. In this case, the antimicrobial activity values were expressed as CFU (dilution 10^−4^ and 10^−6^, three times each). Before the tests were conducted, all the as-electrospun membranes were sterilized with UV light for approximately 20 min.

## 3. Results and Discussion

### 3.1. GOx Encapsulation

Due to their complex three-dimensional structure, size, and homogenous form-factor, most biologically active proteins (such as enzymes) are not easily electrospun into nanofibers. Most often, this issue is overcome by blending enzymes with an appropriate electrospinable polymer. PVA dope prepared in an aqueous solution has been demonstrated to be easily electrospun into nanofibers and used as an appropriate matrix to encapsulate enzymes, such as GOx due to its hydrophilicity, biocompatibility, and non-toxicity [39]. In addition, polymeric microencapsulation is highly beneficial for high protein loading and protection against denaturing agents [40]. As shown in Figure 2, the co-electrospinning of PVA/GOx and PVA/Glu dopes could produce a nanofibrous membrane consisting of fully entangled nanofibers. Since the feed streams of these two dopes were separated from each other, GOx and Glu nanofibers should be obtained as separated nanofibers, unable to react with each other to generate hydrogen peroxide in its dry state. The morphology of the as-prepared membrane was examined using FE-SEM, as shown in Figure 2, two noticeably different kinds of nanofibers could be observed. One is a straight thinner fiber, the other is a larger size fiber with shuttle-shaped beads. As shown in Figure 3a, fibers with a size of 200–400 nm consisted of 45% of the membrane. The other fibers have a much broader size distribution with a range of 400–1800 nm. Due to the complicated three-dimensional structure and strong inter- and intra-molecular forces of GOx, GOx/PVA dope will be electrospun into uneven fibers and often carries some shuttle-shaped beads structure [39]. The thinner size fibers resulted from Glu/PVA dope because PVA dope alone could always be electrospun into smooth fibers with a narrow size distribution. GA vapor crosslinking treatment was expected to prevent the PVA nanofibers from dissolving in water but it affected the morphology of the membrane. As demonstrated in our previous work [38], PVA crosslinked by GA has been confirmed by FTIR analysis. Figure 2b shows that PVA nanofibers curled up slightly and the shuttle-shaped beads on the PVA strings collapsed after GA vapor crosslinking (Appendix A). This possibly occurs due to the high GA concentration (2 M) used for the vapor-crosslinking. The acid-catalyzed GA crosslinking between the PVA chains compacts the three-dimensional network of PVA, which leads to a reduction in free volume between PVA chains for water absorption [41]. Shrinking results from crosslinking were shown by the reducing size of nanofibers appreciably, as shown in Figure 3b, where even though fibers with a size of 200–400 nm still consisted of ca 50% of the membrane, the size distribution was narrowed to 400–1000 nm. Evidently, PVA chains in electrospun nanofibers were indeed crosslinked leading to the shrinkage of nanofibers structure compared to the original ones.

### 3.2. Sustained Release of Hydrogen Peroxide

The release of H_2_O_2_ from the pristine membrane, in which 3 mg/mL GOx and Glu were separately included in PVA dope for simultaneous electrospinning, was first qualitatively detected by colorimetric method using DA-64 as H_2_O_2_-sensitive dye. To handle the very light membrane, the membrane was placed on top of a paper towel which had been casted and dried with DA-64 and HRP mixture solution. Then, the membrane-loaded paper towel was placed on the surface of an agar plate. Moisture from agar would wet both paper towel and membrane so that Glu can be released to react with GOx to produce H_2_O_2_. H_2_O_2_ generated would then react with DA-64 with the help of HRP to produce Bindschedler’s green color [42] as shown in Figure 4. The color intensities developed on agar plates with and without additional supplement of 0.8% *w/v* glucose were compared. Evidently, the PVA-only membrane did not develop any colors on both plates. The GOx membrane, on the other hand, did not produce a green color (Figure 4a bottom right) on the glucose-free agar plate but did develop a green color on the glucose plate. In contrast, the pristine GOx/Glu membranes developed a green color on both agar plates which means H_2_O_2_ could be generated without supplementary Glu.

Quantitatively, generated H_2_O_2_ was also determined using TMB/HRP colorimetric assay due to its rapid visual detection of color development and needlessly sophisticated instrumentation [43]. The assay produced a blue-green colored solution with strong peak absorption at 650 nm (Appendix A Appendix A). A calibration curve of H_2_O_2_ concentration was first established (Appendix A Appendix A) before further experiments were conducted. In order to find the optimal ratio of GOx/Glu encapsulated in the membrane, the GOx amount was fixed at 3 mg/mL in PVA dope. The glucose concentration in the PVA dope was varied from 1 mg/mL to 5 mg/mL. The time courses of H_2_O_2_ released from the simultaneously electrospun pristine membranes into water solution are shown in Figure 5. It seems that the glucose loading concentration was proportional to the amount of H_2_O_2_ generated. After 8 h incubation, ca. 50 μM H_2_O_2_ was released from 1 mg/mL Glu membrane while 150 μM was obtained for 3 mg/mL Glu membranes. With the glucose concentration increased to 5 mg/mL, a faster initial H_2_O_2_ releasing rate was obtained. However, a limiting H_2_O_2_ level ca 150 μM was achieved after 4 h. In other words, 3 and 5 mg/mL Glu membranes possessed a similar H_2_O_2_ generation capacity. Probably, the dissolved oxygen concentration limited the GOx reaction when a higher glucose concentration was employed as in the case of 5 mg/mL Glu membrane [44]. Therefore, 3 mg/mL Glu membrane was chosen for further self-sustained H_2_O_2_ generation studies. In order to release H_2_O_2_ from the membrane and sustain this for a longer time, the pristine membrane was furtherly crosslinked with GA vapor. As expected, as shown in Figure 5, the H_2_O_2_ releasing rate was significantly reduced after crosslinking, approximately 100 µm H_2_O_2_ was obtained after 8 h incubation. After 24 h, the crosslinked membrane still could produce H_2_O_2_ to 160 µM. Evidently, GA vapor crosslinking did not damage GOx activity and the crosslinked PVA structure prevented the membrane from dissolving when in contact with water. The crosslinked PVA chains also slowed down the diffusion of glucose allowing H_2_O_2_ to be produced for a longer time. 

The stability of H_2_O_2_ generation from GOx/Glu membrane stored at room temperature is shown in Figure 6. The membrane of equal size was incubated with an aqueous solution for 4 h before the H_2_O_2_ concentration was measured. The membrane showed a very stable capacity for H_2_O_2_ generation in which approximately the same level of H_2_O_2_ (~140 μM) was obtained after 7 days stored at room temperature. The GA-crosslinked membrane also showed similar stability but with a lower level of H_2_O_2_ (~90 μM) production. Apparently, the lower level of H_2_O_2_ obtained resulted from the GA crosslinking that slowed down the mass transfer rate of glucose, and therefore, reduced the H_2_O_2_ generation.

### 3.3. Antimicrobial Activity

The antimicrobial activity of the pristine GOx/Glu nanofibrous membrane was assessed and showed strong antimicrobial activity against Gram-negative (*Escherichia coli*; BL21) and Gram-positive (*Staphylococcus aureus*; ATCC65389) bacteria as shown by the disk diffusion inhibition assay. As shown in Figure 7, the antimicrobial activity of the GOx/Glu membrane resulted in large inhibition zones for both bacteria. As a reference for the antimicrobial activity of the membrane, 1% of H_2_O_2_ was cast into the PVA-only membrane side by side with the sample membranes. In the normal LB plate which contained no glucose, the GOx/Glu membrane displayed an inhibition zone against *E. coli* much smaller than the reference 1% H_2_O_2_ membrane (34 mm vs 20 mm; Figure 7a). With a supplement of 0.8% *w/v* glucose in the LB plate, the sample membrane demonstrated a similar size of inhibition zone as the reference membrane (32.5 mm vs 30 mm; Figure 7b). This shows that with the supplemented glucose the pristine GOx/Glu membrane has the same antimicrobial activity as 1% H_2_O_2_ against *E. coli*. For the antimicrobial activity assay against Gram-positive *S. aureus*, TSB agar is often used, however, 0.8% *w/v* or more glucose is usually included. In the normal TSB plate containing glucose, the GOx/Glu membrane displayed an inhibition zone with a size similar to that of the 1% H_2_O_2_ reference membrane (Figure 7c). In contrast, the size of the inhibition zone of the GOx/Glu membrane against *S. aureus* was much smaller than that of the reference H_2_O_2_ membrane on the TSB plate without glucose supplement (23 mm vs 37.5 mm; Figure 7d). Again, this demonstrates that the GOx/Glu membrane does possess strong antimicrobial activity against both Gram-positive and Gram-negative bacteria but the supplemental glucose from the surrounding environment could significantly enhance its antimicrobial activity with a capability equivalent to that of 1% H_2_O_2_.

The quantitative analysis of antimicrobial activity was also assessed through viable cell forming units (CFU/mL) reduction. To investigate the antimicrobial sustainability of the membranes, bacteria were cultured under shaking in contact with the membranes for 30 m, 1 h, and 2 h. Glucose of 200 mg/dL in PBS was also supplemented as a culture solution to emphasize the possibility of treating diabetic patients. The initial bacterial concentration was OD = 0.5, the sampled bacteria suspensions were serially diluted in PBS buffer until 10^−4^ and 10^−6^ dilution rate (three times each). As shown in Figure 8, the CFU of both bacteria decreased significantly after contacting the membranes for 30 min. As incubation time increased, the remaining CFU in solution furtherly decreased. It seems that *S. aureus* is more susceptible to H_2_O_2_ to such an extent that nearly no viable cells (CFU) could be detected after 30 min for the pristine membrane with supplemented glucose in solution (Figure 8a). In contrast, appreciable amounts of *E. coli* remained alive after 30 min. As expected, the pristine membrane without GA crosslinking is most effective for killing both bacteria in the presence of supplemented glucose. This is because without crosslinking, GOx would effectively release into the culture suspension with time and react with the supplemented glucose to generate H_2_O_2_ for killing bacteria. As for the GA-crosslinked membrane, even in the environment without supplemented glucose, its antimicrobial activity was still extraordinary in which nearly all viable *S. aureus* were killed after 2 h. However, for the Gram-negative *E. coli* cells, there were still small amounts of viable cells remaining. Supplemented glucose in the bacteria suspensions contributed a significantly enhanced antimicrobial activity of the crosslinked membranes against *S. aureus* (>80% reduction in CFU) as compared with *E. coli* (<20% reduction) for the incubation time of 30 min and 1 h. Gram-positive bacteria have only one cell membrane layer which makes them more vulnerable to antimicrobial agents as compared with Gram-negative bacteria which have an inner and outer cell membrane that afford them better protection against antimicrobial agents. The same antimicrobial trends of these two types of bacteria against GOx-based nanoparticles and direct H_2_O_2_ disinfectants have also been reported previously [45]. 

For the purpose of antimicrobial sustainability tests, pristine GOx/Glu membrane and PVA-only membrane without further sterilization were placed on top of an agar surface. To enhance and confirm the antimicrobial activity, hands were also not washed before handling these membranes. As shown in Figure 9, after 3 days of incubation, microorganisms started to grow on top of the PVA-only membrane while the other two pristine membranes showed no sign of microorganism growth even after 2 weeks of placement. This again showed that the continuous production of H_2_O_2_ from the GOx/Glu membrane can sterilize the membrane itself. Additionally, the sustainability effect of GA crosslinking for the GOx/Glu membrane was also analyzed. Initially, inhibition zones were observed for both pristine and GA-crosslinked membranes. Interestingly, contaminating microorganisms could grow (2 weeks after testing) on the pristine membrane (without GA crosslinking), though possessing a larger initial inhibition zone. In contrast, the GA-crosslinked membrane did not show any contamination and the inhibition zone could still be clearly observed. This shows that the release of in situ-generated H_2_O_2_ from the crosslinked membrane can last for a longer time to protect the membrane from contamination. On the other hand, the pristine membrane could generate H_2_O_2_ faster for bacterial killing but once H_2_O_2_ was exhausted it could not protect itself from bacteria contamination.

In detail, the antimicrobial activity of the as-prepared nanofibrous membrane was mainly due to cytotoxicity of in situ reaction products of GOx catalyzed glucose oxidation. In this nanofibrous membrane antimicrobial system, glucose can be offered either in situ or supplemented externally. Glucose supplemented from the surrounding can enhance H_2_O_2_ generation from the system. Therefore, the GOx/Glu membrane would be useful not only for a normal wound dressing but also a more potent antimicrobial wound dressing for diabetic patients who usually have a higher level of blood sugar. The H_2_O_2_ generated from the membrane will act as a reactive oxygen species (ROS), producing hydroxyl free radicals (•OH), attacking crucial cellular constituents, such as proteins, lipids, and nucleic acids, and thereby killing the bacterial cells through peroxidation and disruption of cell membranes.

## 4. Conclusions

A self-sustained hydrogen peroxide-generating membrane was successfully pre-pared by separately encapsulating GOx and glucose in PVA nanofibers via simultaneous electrospinning of PVA/GOx and PVA/Glu dope. GA vapor cross-linking made the membrane water-resistance and slowed down its H_2_O_2_ releasing rate. The H_2_O_2_ production level in an aqueous solution could be well maintained even after 7 days of incubation of the dry membrane at room temperature. Because of its extremely high specific surface area, PVA nanofibrous membrane could respond swiftly to an aqueous environment, and as a result, glucose was released from PVA nanofibers and reacted with encapsulated GOx to generate H_2_O_2_. The sustained release of H_2_O_2_ from the membrane was shown to possess strong antimicrobial activity against Gram(−) *E. coli* and Gram(+) *S. aureus*, with and without the presence of glucose in their environment. In addition, the sustained release of H_2_O_2_ also demonstrated that the as-prepared membrane was self-sterilizable and that most microorganisms contaminated on the membrane could be effectively killed continuously. This GOx/Glu membrane may find potential application as versatile antimicrobial materials in the field of biomedical, food, and health industries, especially in facing challenges relating to wound healing in diabetic patients.

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
