# Peer review of "Nanofibrous Membrane with Encapsulated Glucose Oxidase for Self-Sustained Antimicrobial Applications"

_membranes, 2021, doi:10.3390/membranes11120997_

Round 1

Reviewer 1 Report

In this paper, the authors report the development of polyvinyl alcohol (PVA) nanofibrous membranes with encapsulated glucose oxidase (GOx) and glucose (Glu) for versatile antimicrobial applications. This paper reports interesting results, and the results are supported by sufficient data, so I think that this paper can be published in membranes after minor revisions.

My detailed comments are as follows.

1) An error bar should be displayed on the graph in Figure 3.

2) Data that can confirm the crosslinking of PVA by GA should be presented. (e.g. FT-IR spectra)

3) (P 5: L 193-194) The reason why crosslinking by GA reduces the size of nanofibers should be clearly explained.

4) (P 7: L 245-247) Please, add the data of dissolved oxygen concentration over time in the revised manuscript.

5) (P 9) Figure 5 -> Figure 6 (in the figure caption)

6) (P 9) The reason why the H2O2 concentration levels of the GA crosslinked samples are different from each other in Figure 5 and  Figure 6 should be clearly explained.

Author Response

Thank you for your detailed review. The responses to your comments had been attached below.

Please see the attachment and also revised manuscript.

Reviewer 2 Report

This paper reports Nanofibrous Membrane with Encapsulated Glucose Oxidase for Self-Sustained Antimicrobial Applications. The study was well organized and the report was well written. However, one aspect needs to be included in the report. Considering the sensitivity of an enzyme, the authors must consider its stability upon storage or exposure to unfavorable conditions. Another aspect that needs to be enhanced is the statement of novelties. The authors do not provide sufficient literature review to show the research gap and the novelty of the research. Tons of publications are available on nanofiber as the wound dressing but with minimum elaboration in this work.

Author Response

Thank you for your detailed review.

Please see attachment for the responses to your comments.

Reviewer 3 Report

It is a very good study with overall adequate presentation of experimental results. Some additions are needed:

1) Authors should further emphasize on the novelty of their work.

2) Some minor typos, grammar and syntax errors should be carefully revised and corrected accordingly.

3) Reference can be even more updated (more recent relative works).

4) The discussion section must be enhanced with more comparisons and critical approach.

Author Response

Thank you for your review comments and suggestions.

The responses to these comments are attached. Thank you.

Round 2

Reviewer 3 Report

All my comments of the initial submission have been correctly replied and included in the revised manuscript. The quality of this work has been drastically improved after revision and therefore I recommend its publication as it is.